# 5G-NR Physical Layer-Based Solutions to Support High Mobility in 6G Non-Terrestrial Networks

Chaitali J. Pawase 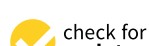 and KyungHi Chang *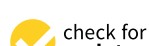

Department of Electrical and Computer Engineering, Inha University, 100 Inha-ro, Michuhol-gu, Incheon 22212, Republic of Korea
* Correspondence: khchang@inha.ac.kr

**Abstract:** Non-terrestrial network (NTN) systems can offer wide area coverage for applications requiring high mobility, which is expected in the sixth generation (6G) of telecommunication systems. This paper proposes a high-mobility support system based on the 5G-NR physical layer components for NTN connectivity. In this paper, we propose the optimization of 5G-NR numerologies and the impact of various modulation and coding schemes (MCS), 3GPP NR-NTN channel models, and MIMO/beamforming schemes with link-level simulation under pilot-aided-based perfect and DM-RS-based practical channel estimation at stationary UE and high mobility of 500 km/h, respectively. This paper also develops a link-level simulation of the 5G-NR physical downlink shared channel (PDSCH) under the 3GPP NR-NTN tapped delay line (TDL) channel model to support UE mobility up to 500 km/h. The bit error rate (BER), maximum achievable throughput (Mbps), and spectral efficiency (bps/Hz) are analyzed for the 5G-NR-based potential elements to be utilized in the evolution of NTN. Furthermore, the denser DM-RS symbol pattern is proposed for utilization in channel estimation to support high mobility, as simulation results prove their capability of fast decoding while using the front-loaded symbol structure. The simulation results show that the large 5G-NR numerologies, such as 120 kHz and DM-RS-based channel estimation, support the high UE mobility by providing high link reliability and the maximum achievable throughput of 368.832 Mbps and spectral efficiency of 3.68 bps/Hz under 64-QAM for TDL-E (LOS) channel model, which can also be a potential solution to support transonic speed mobility in the NTN of 6G services.

**Keywords:** 3GPP; 5G-NR; 6G; beamforming; channel model; DM-RS; MIMO; non-terrestrial networks (NTN); numerology

## 1. Introduction

In 6G networks, 5G technology is expected to support NTN with enhanced terrestrial coverage and provide upgraded services [1]. Air-to-ground networks, high-altitude platforms (HAPs), and satellite communication networks are all part of the NTN. NTN communications may become critical for 6G wireless to ensure enormous flexibility and the convergence of terrestrial and satellite networks [2]. Air-to-ground networks deploy ground stations that serve the same functions as base stations (BSs) in terrestrial mobile networks to ensure service consistency in order to provide airplanes with in-flight connections [3,4] and to improve sweep coverage [5,6]. Ground station antennas in an air-to-ground network are pointed upward, and ground station inter-site distances are substantially greater than in terrestrial mobile networks. Because of its inherent compatibility with air-to-ground networks, 3GPP NTN research has thus far focused primarily on satellite communications networks [7]. The third-generation partnership project (3GPP) is now working to standardize 5G new radio (NR) [8,9]. New features and solutions, including service-based network architecture, two-layer (central unit and distributed unit) radio access network architecture, and air interface with flexible frame structure, have been employed in 3GPP NR technical reports (TR) and specifications (TS) [1,10,11]. The work item for 3GPP Rel-17 was established to define and assess solutions

in the field of NTN for NR, with a priority on satellite access [11]. Furthermore, the Rel-18 and Rel-19 study items have been recognized, indicating ongoing research.

The physical downlink shared channel used by the 5G new radio to transmit user data is known as PDSCH. The reference signals for 5G-NR, namely the demodulation reference signal (DM-RS) and phase tracking reference signal (PT-RS), are related to PDSCH. According to TS 38.211 [8], these signals are generated inside the PDSCH allocation. In coherent PDSCH demodulation, the channel estimation is carried out by DM-RS. The PT-RS is implemented in 3GPP 5G NR to address common phase error (CPE). The phase noise produced by local oscillators causes significant degradation in system performance. The inter-carrier interference (ICI) and CPE interference are both caused by it. With CPE, each subcarrier's rotation of a received symbol is identical. Subcarriers lose orthogonality because of ICI. The main function of PT-RS is to evaluate and reduce the impact of CPE on system performance.

We contribute to the research on NR-based solutions to support transonic speed in the 6G-NTN through this paper. This paper is motivated by the interest that industry and academics have shown in NTN networks over the last decade. It serves as a key identifier and emphasizes the value added for 5G networks. The growing interest in NR-NTN has led to the publication of numerous surveys and tutorials. Nevertheless, the details of 3GPP standardization are not addressed. Instead, the paper briefly discusses the role of NTNs, application scenarios, and networking challenges. The References provide in-depth coverage of the architectural and key technical concerns for 5G systems, including satellites; however, the discussion there is mostly centered on radio access network (RAN) issues. It is especially important to mention the work conducted by the 3GPP Service, System Aspects (SA), and Core and Terminals (CT) working groups on the NTN support for high mobility. It is important to verify which set of 5G-NR physical layer specifications will support the high UE mobility communication in NTN must be determined. Our main contributions are as follows:

- To better comprehend the function of NTN within the 5G new radio (NR) system, this paper introduces the NTN wireless system, highlights its key features based on 3GPP technical reports, and discusses the current state of the art on NTN.
- This paper summarizes the work the 3GPP is currently conducting to enable NTN as a component of NR technology and to identify remaining issues and potential research directions.
- With these considerations, the 5G-NR numerologies are optimized to support high mobility.
- We analyze the link-level performance of a 5G-NR physical channel for downlink user data transmission, known as a physical downlink shared channel (PDSCH) under a 3GPP NR-NTN channel model, such as single-path Rayleigh, Rician, and multi-path tapped delay line (TDL) channel models, in terms of the bit error rate (BER), maximum achievable throughput, and spectral efficiency.
- We identify the impact of various numerology and modulation and coding schemes (MCSs), pilot-aided perfect and DM-RS-based practical channel estimation, and MIMO/beamforming schemes at the stationary UE mobility and 500 km/h.
- Based on the link-level simulation results, we propose 5G-NR-based potential solutions for NTN to support high mobility in the 6G technology evolution.

The structure of this paper is as follows. Section 2 presents a literature review of related work in NTN. Section 3 provides an overview of NTN with its structure in 6G, scenarios, use cases, major challenges and solutions, and the channel models proposed by 3GPP to support NR-NTN. In Section 4, 5G-NR-based solutions are proposed for high mobility. These solutions are analyzed using the link-level simulation in Section 5, where the simulation results and analysis are discussed. The conclusions are given in Section 6.

## 2. Related Works

The NR-NTN research is not limited to 3GPP reports. Babich et al. presented a novel network architecture for an integrated nanosatellite-5G system operating in the

millimeter wave (mmWave) domain [12], whereas [13] identified the most promising satellite networking configurations and discussed some design trade-offs in this domain. The authors in [14–16] describe the physical layer communication research beyond 5G because of the numerous extra features provided to NR, PHY abstraction of NR-based systems is a difficult task. The variable number of resource blocks (RBs) and OFDM symbols that can be allocated to each user in NR results in a significant increase in the number of supported transport block sizes (TBSs) [15]. The introduction of LDPC coding with varying lifting sizes and two different base graph types complicates the code block segmentation procedure at PHY [16]. Furthermore, NR supports modulation orders up to 256-QAM and a wide range of MCS table combinations, which is discussed in 6G evolution [17].

Moreover, 6G researchers are working on constructing non-terrestrial networks (NTN) to enable ubiquitous and high-capacity global connections [18]. Unlike previous wireless generation networks, which have traditionally been designed to provide connectivity for a quasi-bi dimensional space, 6G envisions a three-dimensional (3D) heterogeneous architecture in which terrestrial infrastructures are supplemented by non-terrestrial stations such as unmanned aerial vehicles (UAVs), high-altitude platforms (HAPs), and satellites [19]. These features can provide on-demand, cost-effective coverage in densely populated and unserved areas and ensure backhauling, support for high-speed mobility, and high-throughput hybrid multiple services. In particular, the significance of NTN has been acknowledged in standard activities [20].

Marco Giordani et al. [2] reviewed NTN research activities, described the features and enabling technologies of NTNs in the 6G landscape, and focused on the problems in the field that remain open for research. For example, they examined how different networking configurations affect the performance of an NTN scenario in which aerial/space vehicles use millimeter wave (mmWave) frequencies to offer access connectivity to on-the-ground mobile terminals. The proposed DM-RS design sets by Gosan Noh et al. [21] are intended to provide accurate channel estimates in a very high-mobility environment of up to 500 km/h while limiting the DM-RS overhead to an acceptable degree in the terrestrial networks. Extensive link-level simulations from [22] show that even with significant Doppler shift/spread, the proposed DM-RS design sets can achieve satisfactory block error rate (BLER) and spectrum efficiency performance. DM-RS patterns applicable to the 3GPP 5G-NR high-speed train (HST) scenario with MIMO schemes and TDL channel model are also discussed in [21,22]. Fumihiro Hasegawa et al. discussed initial access, mobility management, and linear cell designs in [23], in which they discuss the evaluation parameters for the 3GPP-adopted HST scenario and the proposed physical layer technologies for HST applications, including numerology and reference signal designs. For high-speed rail broadband mobile communications, the authors in [24] propose massive MIMO and beamforming-based methods that support high mobility. The analysis of flexible numerologies of 5G-NR to support integrated-satellite-terrestrial networks with low earth orbit (LEO) satellites are described by authors in [25].

Furthermore, there are multiple case studies of NTN deployments in other countries, as well as efforts by international organizations and NGOs for NR-NTN evolution in 6G [7]. On the other hand, despite recent research, some questions remain unanswered for proper network architecture [26]. While some authors focused on independent aerial/space architectures, there has yet to be a formalization of the constraints and opportunities associated with a multi-layered network in which various non-terrestrial stations interact at different altitudes in an integrative manner [27]. Other papers, [19,26], discussed how to improve the protocol stack design in a space–air–ground integrated network but did not comprehensively investigate the most recent technology breakthroughs to accomplish performance optimization. Furthermore, a thorough description is given for NTN supporting technological solutions, and research goals are now dispersed throughout multiple technical studies, making them difficult to grasp without the necessary background [4,28]. All these characteristics must be considered when simulating the NR performance. The evolution of 6G-non-terrestrial networks is examined in this paper. We analyze the 5G-NR physical layer link-level performance to support high UE mobility up to 500 km/h to propose NR-based solutions for NTN. Table 1 briefly summarizes the research work conducted by the 3GPP groups and other researchers.

**Table 1.** A summary of related works.

| Reference | Brief Description | Reference | Brief Description |
|---|---|---|---|
| 3GPP TR 38.811 (Release-15) [1] | <ul><li>Study on NR to support non-terrestrial networks</li><li>Study channel model, deployment scenarios, and potential key impact areas</li></ul> | Calvanese S. E. et al. [17] | <ul><li>NR supports modulation orders up to 256-QAM</li></ul> |
| Giordani M. et al. [2] | <ul><li>Reviewed NTN research activities</li><li>Described the features and enabling technologies of NTNs in the 6G landscape</li><li>Focused on the problems in the field that remain open for research. For example, they examined how different networking configurations affect the performance of an NTN scenario in which aerial/space vehicles use millimeter wave (mmWave) frequencies to offer access connectivity to on-the-ground mobile terminals</li></ul> | Liberg O. et al. [19], Rumney M. [27] | <ul><li>Discussed how to improve the protocol stack design in a space–air–ground integrated network but did not comprehensively investigate the most recent technology breakthroughs to accomplish performance optimization</li></ul> |
| Kim J. et al. [7] | <ul><li>Multiple case studies of NTN deployments in other countries and efforts by international organizations and NGOs for NR-NTN evolution in 6G are studied</li></ul> | Noh G. et al. [21,22] | <ul><li>DM-RS design sets are intended to provide accurate channel estimates in a very high-mobility environment of up to 500 km/h while limiting the DM-RS overhead to an acceptable degree in the terrestrial networks</li><li>Link-level simulations show that with significant Doppler shift/spread, the proposed DM-RS design sets can achieve satisfactory block error rate (BLER) and spectrum efficiency performance</li><li>DM-RS patterns applicable to the 3GPP 5G-NR high-speed train scenario with MIMO schemes and TDL channel model</li></ul> |
| 3GPP TR 38.821 (Release 16) [10] | <ul><li>Study on solutions for NR to support non-terrestrial networks</li><li>Study a set of necessary features enabling NR support for NTN</li></ul> | Hasegawa F. et al. [23] | <ul><li>Initial access, mobility management, and linear cell designs in which they discuss the evaluation parameters for the 3GPP-adopted HST scenario</li><li>Proposed physical layer technologies for HST applications, including numerology and reference signal designs</li></ul> |

**Table 1.** *Cont.*

| Reference | Brief Description | Reference | Brief Description |
|---|---|---|---|
| Babich et al. [12] | • Presented a novel network architecture for an integrated nanosatellite-5G system operating in the millimeter wave (mmWave) domain | Ge Y. et al. [24] | • Proposed massive MIMO and beamforming-based methods that support high mobility for high-speed rail broadband mobile communications |
| Jiang H. et al. [13] | • Identified the most promising satellite networking configurations and discussed some design trade-offs in this domain | Jayaprakash A. et al. [25] | • The analysis of flexible numerologies of 5G-NR to support integrated satellite-terrestrial networks with low earth orbit (LEO) satellites are described |
| Raghavan V. et al. [14] | • Describe the physical layer communication research beyond 5G because of the numerous extra features provided to NR, PHY abstraction of NR-based systems is a difficult task | Kim J. H. et al. [28] | • Thorough description is given for NTN supporting technological solutions, and research goals are now dispersed throughout multiple technical studies, making them difficult to grasp without the necessary background |
| Khalid W. et al. [15] | • The variable number of resource blocks (RBs) and OFDM symbols that can be allocated to each user in NR results in a significant increase in the number of supported transport block sizes (TBSs) | 3GPP TR 22.822 (Release 16) [29] | • Study on using satellite access in 5G<br>• Identify use cases and the associated requirements |
| Our Contributions | • Introduces the NTN wireless system, highlights its key features based on 3GPP TR and TS<br>• 5G-NR numerologies are optimized to support high mobility based on the link-level performance of a 5G-NR physical channel for downlink user data transmission, which is a PDSCH under a 3GPP NR-NTN channel model, such as single-path Rayleigh, Rician, and multi-path tapped delay line (TDL) channel model in terms of BER, maximum achievable throughput, and spectral efficiency<br>• Identify the impact of various numerology and MCS levels, pilot-aided perfect and DM-RS-based practical channel estimation, and MIMO/beamforming schemes at the stationary UE mobility and 500 km/h<br>• Proposes 5G-NR-based potential solutions for NTN to support high mobility in the 6G technology evolution | | |

## 3. Overview of Non-Terrestrial Networks

The 3GPP TR 38.811 [1] describes the deployment scenarios, channel models for NTN, and the regions with significant NR interface effects. This TR also specifies the NTN deployment scenarios and system features (such as architecture, altitude, and orbit). TR 38.821 [10] addresses a variety of prerequisites and adjustments that allow the NR protocol to function in NTN, with a focus on satellite access. A network based on UAS and including HAPs may be a remarkable example of non-terrestrial access with a lower delay/Doppler value and fluctuation rate [2]. The objectives of this initiative include performance evaluation of 5G-NR in a few deployment scenarios (LEO satellite), gathering potential effects on the physical layer, defining solutions for them, solutions for 5G-NR associated with layers 2 and 3, solutions for the RAN architecture, and related technologies [30]. The 3GPP TR 22.822 [29] standard enables service continuity between terrestrial NG-RAN and NTN-based NG-RAN owned or covered by an operator agreement. This TR aims to define service delivery use cases for addressing NTN-based access component integration into the 5G system, as well as new services and requirements (i.e., setup, configuration, maintenance, and regulation).

The 3GPP TR 28.808 [30] describes the major challenges of a 5G network with integrated satellite components, such as business roles, services, management, and orchestration. This scenario investigates potential solutions, aspires to simplify the integration of satellite technology into present business models, and considers the management and orchestration parts of current 5G networks. Adapting 5G-NR for satellite communications was considered in [10] based on Release 15 of the NR specifications. The authors in [10,11] emphasized the issues with connected mode and idle mode mobility and NR-specific network design components in both LEO and GEO-based NTN systems, focusing on the physical layer and user plane characteristics. The network, or elements of a network, that have a base station or transmission equipment relay node as a payload on board an airborne or spaceborne spacecraft are referred to as non-terrestrial networks. Satellites are classified as spaceborne vehicles based on their orbital heights [2], which range from 400 km to 2000 km for low earth orbit (LEO), 2000 km to 35,786 km for medium earth orbit (MEO), and 35,786 km for geostationary earth orbit (GEO). GEO [3] are geosynchronous orbits with a zero angular inclination to the equator. The non-terrestrial network scenarios and use cases are shown in Figure 1. The range of terrestrial and non-terrestrial networks is shown in Figure 1, along with the different colored vectors for the different use cases and their communication between the core network to satellite, HAPSs/UAVs, and drones with the terrestrial networks, such as high-speed trains, as a high-mobility application scenario as well as communication between rural, urban, and remote areas and the NTN.

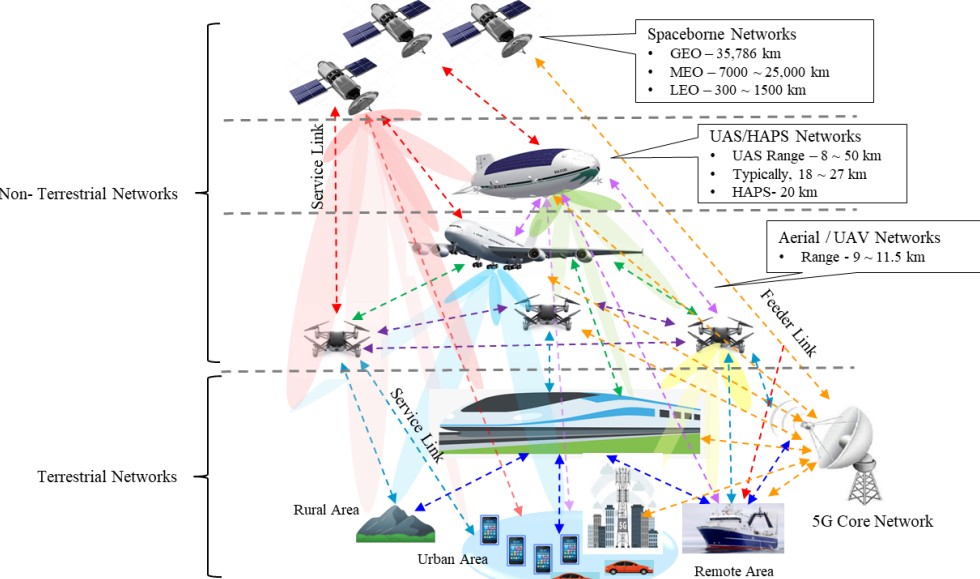

**Figure 1.** Non-terrestrial network scenario and use cases.

### 3.1. NTN in 6G Era

A terrestrial terminal, an aerial or space station that may function similarly to a terrestrial base station, a service link between the terrestrial terminal, aerial or space station, and a gateway that joins the non-terrestrial access network to the core network via a feeder link are the main components of non-terrestrial networks in 6G technology [2]. For NTN development in the 6G era, satellite stations, HAPs, and unmanned aerial vehicles (UAVs) can be taken into consideration. Several NTN architectures are being promoted by the 3GPP depending on how integrated the various airborne and spaceborne parts are [18]. The transparent satellite-based Radio Access Network (RAN) architecture is precisely what the 3GPP specifically expects, in which the satellite transfers the user's signal from the feeder link to the service link and vice versa. The regenerative satellite-based RAN design, which allows for inter-satellite communication while implementing signal regeneration in the satellite payload, will be taken into consideration for 6G [4]. Moreover, 6G technology will offer multi-connectivity architecture with two transparent RANs (either GEO or LEO or a combination of both), which integrates terrestrial and non-terrestrial access.

Non-terrestrial technology has long been thought to support operations such as home delivery, weather forecasting, video surveillance, television transmission, remote sensing, and navigation. To enable more advanced use cases, such as Communication Resilience and Service Continuity, Global Satellite Overlay, Ubiquitous Internet of Things (IoT) Broadcasting, Advanced Backhauling, and Energy-Efficient Hybrid Multiplay, the recent technological advancements in the aerial/space industry have however made it possible for integration between terrestrial and non-terrestrial technologies. The few enabling technologies to support NTN in 6G for architecture, spectrum, antenna, and higher layers advancements, respectively, include gallium nitride (GaN), cognitive spectrum, multi-beam structures, and TCP spoofing and multiplexing [2].

### 3.2. NTN Scenario

The user control connection connects the satellites directly to the on-ground handheld user equipment (UE) via the new radio (NR) air interface [4]. The top architecture denotes a solution with a transparent payload, whereas the bottom denotes a solution with a regenerative payload. The gNB is located conceptually at the gateway (GW) in the first scenario, but it is realized on board with regenerative payloads. In both scenarios, the user access link between the satellite and the on-ground UE is implemented using the classic NR-Uu air interface [30]. The position of the gNB determines the solution for the feeder link between the satellite and the GW. A transparent payload serves as a radio-frequency repeater, necessitating the employment of an NR-Uu solution for the feeder connection. In contrast, the gNB is located on the satellite, and the feeder link is implemented via the air interface between a gNB and the next-generation core (NGC) network. This significantly reduces propagation delays for NR PHY and MAC operations, easing the changes required for NTN [31].

### 3.3. NTN Use Cases

Non-terrestrial systems have been proposed to enable several applications, including weather forecasting, video surveillance, TV broadcast, remote sensing, and navigation, for several years. On the other hand, recent technological advances in the aerial/space sector have enabled the implementation of more sophisticated use cases, such as distributed computing and content broadcasting, service boosting for users in congested areas, eMBB in underserved areas, and multi-connectivity for service continuity via cellular networks [2,4].

### 3.4. NTN Challenges and Solutions

The propagation delay for GEO satellites in NTN networks can be hundreds of milliseconds because of the incredibly large gap between the gNB and the UE, particularly in bent pipe configurations [2]. The UE Doppler effect and the rapid relative motion of spaceborne objects, such as LEO satellites, pose significant technological hurdles for NTN in achieving

high mobility, such as transonic speed mobility of around 1000 km/h [1,2]. For a 2 GHz carrier frequency, there is a Doppler change of more than 7 km/s and more than 20 ppm [4,19,20]. The Doppler effect and transmission delays are not intended to be considered by current NR standards, which are primarily intended for cellular systems.

Directionality is required while operating at mmWaves to achieve an adequate link budget and maintain high-capacity connections [24]. Fine beam alignment substantially influences the design of control activities in this situation, including user tracking, handover, and radio link failure recovery [22]. Because of the speed of aerial and space platforms, these issues are more critical in the extra-terrestrial domain because beam alignment may be lost before a data transfer is completed. The channel may become non-reciprocal at high speeds owing to the higher Doppler experience [23], resulting in less feedback than a broadcast channel [29]. Therefore, NR-NTN is critical for 3D spatial mobile connectivity investigations [13].

The physical layer management procedure and scheduling concerns are the primary technical issues that must be resolved [25]. As a result, power control, AMC/CSI feedback, beam switching, and uplink transmission time can all be used to overcome difficulties with the physical layer control technique [14]. While timing issues in 6G-NTN can be addressed through downlink and uplink synchronization, the new physical random-access channel (PRACH) protocol for the RACH process, and HARQ for improved mobility [15,16].

### 3.5. Channel Models for NR-NTN

The third-generation partnership project (3GPP) released a technical report on channel models for frequencies ranging from 0.5 to 100 GHz [9] that would be employed in 5G communication systems to support the development of 6G technology services. The objective of this paper is to contribute to the design and analysis of physical layer performance for non-terrestrial networks using single- and multi-path channel models under high UE mobility. In 3GPP technical reports, two channel models, such as clustered delay line (CDL) and tapped delay line (TDL), have been proposed to be utilized in link-level simulations [1,8–10].

The CDL is a channel model used when the received signal consists of several distinct delayed clusters. Each cluster consists of many nodes in the network with the same delay but varying angles of departure (AOD) and angles of arrival (AOA). CDL channel models expand TDL profiles and are developed for 3D channels. There are five different CDL profiles available. There are three CDL models applicable to NLOS: CDL-A, CDL-B, and CDL-C. CDL-D and CDL-E are CDL models for the LOS environments [9]. The generation of the azimuth angles of departure and arrival is the first stage in creating the channel coefficients for the CDL profiles. Therefore, the arrival angles for the $m$th ray of the $n$th cluster can be determined as [8,9]:

$$\varnothing_{n,m}, AoA = \varnothing_n, AoA + C_{ASA}\,\alpha_m \tag{1}$$

The azimuth angle of arrival ($AoA$) is defined by $\varnothing_n, AoA$, and the azimuth spread of arrival angle is represented by $C_{ASA}$. The ray offset angle within a cluster is defined by $\alpha_m$. Similarly, the departure angles for the $m$th ray of the $n$th cluster can be determined as [8,9]:

$$\varnothing_{n,m}, AoD = \varnothing_n, AoD + C_{ASD}\,\alpha_m \tag{2}$$

The azimuth angle of departure (AoD) is represented by $\varnothing_n, AoD$, and the cluster-wise Root Mean Square (RMS) azimuth spread of the departure angle is defined by $C_{ASD}$.

The TDL channel model offers five different profiles depending on the environment scenario. Present profiles for non-line-of-sight (NLOS) scenarios are TDL-A, TDL-B, and TDL-C, whereas TDL-D and TDL-E are the corresponding LOS profiles [9]. The channel impulse response of the TDL channel model is specified as:

$$H(t,\tau) = \sum_{k=1}^{n} a_k(t)\delta(\tau - \tau_k) \tag{3}$$

The amplitude at the $\tau_k$ delay for the $k$th tap is denoted as $a_k(t)$. Except for the TDL-C profile, which has 24 taps, all designed NLOS profiles contain 23 taps. The TDL-D and TDL-E are LOS profiles containing 13 and 14 taps, respectively, with a Ricean fading distribution for the first tap. The parameters for the normalized latency of each profile and the relative power for each tap can be found in [8,10]. Doppler spectra for each tap are characterized by a classical (Jakes) spectrum shape and a maximum Doppler shift $f_D$.

$$f_D = |\overline{v}|/\lambda_0 \tag{4}$$

The Doppler spectrum for those taps also contains an amplitude peak, resulting in a fading distribution with the required K-factor at the Doppler shift $f_s = 0.7 f_D$ [10].

In order to select the propagation characteristics for a high-mobility scenario, we have utilized the tapped delay line (TDL) channel model developed by 3GPP in [1,9] for our link-level simulation. Since our aim is to identify the impact of the different DMRS symbol patterns and the MIMO configurations on the high UE mobility, we have utilized the TDL channel model. In [8,9,21,22] TDL-A for NLOS and TDL-E for LOS models are recommended for the evolution of DMRS of the 3GPP 5G-NR high-speed train scenario, where the Rician K-factor is 13.3 dB. In these profiles, the TDL channel offers the delay spread as a parameter to scale the path delays, which is a scaling delay spread of 10 ns. Using a shorter delay spread for LOS applications and longer delay spreads for NLOS paths is recommended by [9]. The power delay profiles (PDPs) of selected TDL-A and TDL-E are given in [1,9]. The tap amplitudes follow Rayleigh distributions, and in line-of-sight cases, first tap amplitudes follow Rician distributions; hence, from these PDPs, the one tap of Rayleigh and Rician are used as a single-path channel to analyze link-level performance.

## 4. Proposed 5G-NR-Based Solutions for High Mobility

This paper proposes 5G-NR physical layer components for the downlink communication to meet high-mobility requirements in NTN. The flexible numerology is one of the essential features of the 5G-NR physical layer. Figure 2 shows the NR frame structure specified in 3GPP TS 38.211 [19] for flexible numerology. The minimum subcarrier spacing is determined by the carrier frequency, phase noise, and Doppler effect [25]. A smaller subcarrier spacing would generate either high error vector magnitude (EVM) owing to phase noise or very high requirements on the local oscillator [32].

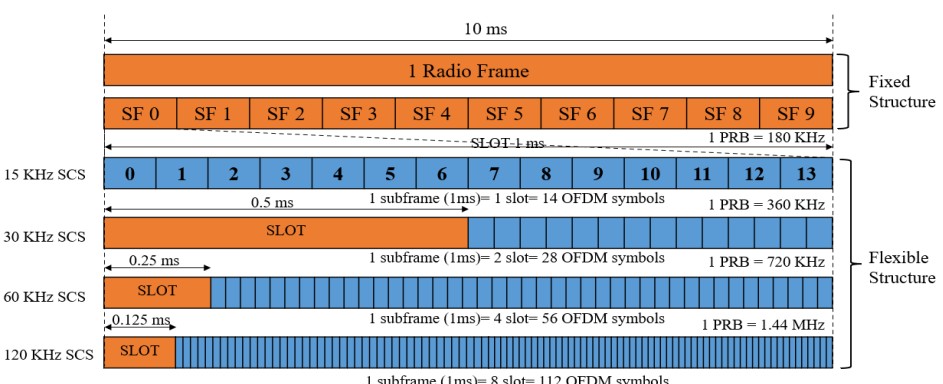

**Figure 2.** 5G-NR frame structure with flexible numerology.

In high-mobility scenarios, excessively narrow subcarrier spacing can degrade the performance. The required cyclic prefix (CP) overhead and projected delay spread establish an upper limit for subcarrier spacing. Therefore, choosing a very high subcarrier spacing may result in high CP overhead. The channel bandwidth is determined by the OFDM modulator's maximum fast Fourier transform (FFT) size and subcarrier spacing [32]. Based on these factors, the subcarrier spacing should be robust to phase noise and the Doppler effect because of the high mobility and provide the appropriate channel bandwidth. In

this paper, we propose adaptive numerology to support UE mobility of 500 km/h for NTN based on the optimization of sub-carrier spacing.

The PDSCH is used to downlink user data, system information, paging, and some higher-layer information regarding user equipment to all connected devices within the covered area [8]. It is the primary data transmission channel in 5G-NR allocated for the users on an active and adaptable basis. The processing chain for link-level simulation is shown in Figure 3. In this processing chain, we configure the PDSCH transmitter and receiver in which the basic communication steps are followed by generating the data, PDSCH encoding, MIMO precoding, modulation, propagation channels such as the TDL channel model and noise addition, demodulation, pilot-aided perfect and DM-RS-based practical channel estimation and compensation, and LDPC decoding to obtain the BER, maximum achievable throughput, and spectral efficiency. The modulation schemes, such as QPSK and different orders of QAM (e.g., 16, 64, and 256), are used in NR-PDSCH. For NR data channels, a low-density parity-check (LDPC) coding scheme is used. The TDL channel model is utilized for the link-level performance analysis of the simulation. The single-path (1P) Rayleigh and Rician fading components are analyzed from the TDL channel model, designed by 3GPP, where one tap is considered a single path [10]. Table 2 lists the simulation parameters for link-level performance analysis. This paper proposes solutions for high mobility based on the optimization of numerologies and the impact of various modulation and coding schemes (MCS), 3GPP NR-NTN channel models, and MIMO/beamforming schemes with link-level simulation under pilot-aided-based perfect and DM-RS-based practical channel estimation at stationary UE and high mobility of 500 km/h.

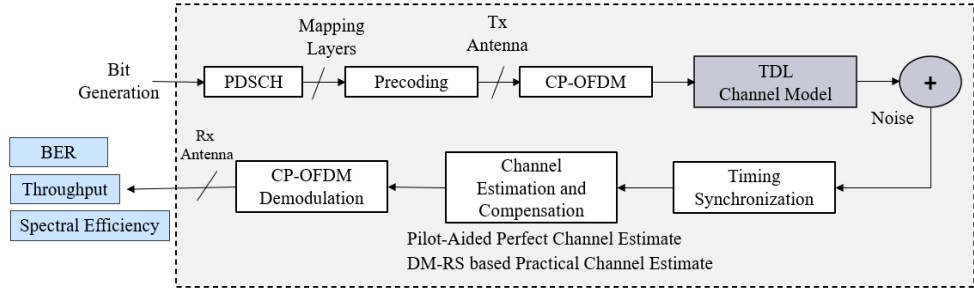

**Figure 3.** Link-level simulation model of the 5G-NR physical layer to support high mobility.

**Table 2.** Link-level simulation parameters.

| Parameters | Values |
|---|---|
| Carrier Frequency | 5.9 GHz |
| System Bandwidth | 100 MHz |
| Sub-Carrier Spacing (SCS) | 15, 30, 60, 120 kHz |
| Cyclic Prefix | Normal |
| Modulation Scheme | QPSK,16,64,256-QAM |
| Propagation Channel Model | 1-P Rayleigh, 1-P Rician, TDL-A, and TDL-E |
| Coding Rate | 490/1024 LDPC |
| Transmitting Antenna (Tx) | 1/2 |
| Receiving Antenna (Rx) | 1/2/4/8 |
| Channel Estimation | Perfect and Practical (DM-RS + PTRS) |
| UE Mobility | Stationary UE and 500 km/h |
| Beamforming Technique | Precoding |

Channel equalization is one of the most challenging issues in wireless communication because signal propagation is affected by various factors, including path loss, phase shifts, delay spread, fading, and Doppler spread. The latter is particularly problematic in mobile communication systems because it can alter the frequency of the transmitted signal significantly on the receiver side. This paper analyzes the impact of the channel estimation method on high mobility. The pilot-aided perfect channel estimation and DM-RS-based practical channel estimation [21] are used in this paper. The transmission of pilot tones within the OFDM symbols with previously determined amplitudes and phases at the receiver side is one of the techniques used in OFDM systems for channel estimation. The data and the pilot subcarriers will have identical amplitude and phase variations while transmitting the composite OFDM signal on the channel. On the other hand, it is possible to estimate the channel characteristics in the frequency domain at these pilot frequencies because of prior knowledge about pilot levels prior to the advent of the channel effect. Understanding some aspects of the channel transfer function makes it easier to estimate the remainder of the transfer function at the desired data sub-carrier frequencies, which greatly reduces the channel effect and improves reception quality.

The 3GPP TS 38.211 [19] specifies the mapping types A (slot-wise) and B (non-slot-wise) for PDSCH symbol allocation. The first DM-RS symbol for mapping type A can be found at position 2 or 3 in a slot, which is denoted as $I_0$. In contrast, the duration of OFDM symbols is denoted as $I_d$ in Figure 4, which is defined as the number of symbols between the first and last OFDM symbol of the slot. Fast decoding is possible because of the front-loaded DM-RS, allotted in the first position of the slot [18]. This paper suggests using front-loaded DM-RS-based channel estimation with a denser DM-RS symbol pattern to support high mobility in NR-NTN evolution. A significant performance impact will be experienced if only a front-loaded DM-RS is allocated, especially in high-mobility applications where the channel is sensitive to time-domain oscillations [18]. As a result, it is crucial to allocate more DM-RS symbols after the front-loaded symbols. As a result, the use of more DM-RS symbols is suggested to produce a denser DM-RS pattern, as shown in Figure 4. As a result, increasing time-domain DM-RS density can also result in an increase in DM-RS overhead and degradation in spectral efficiency [18].

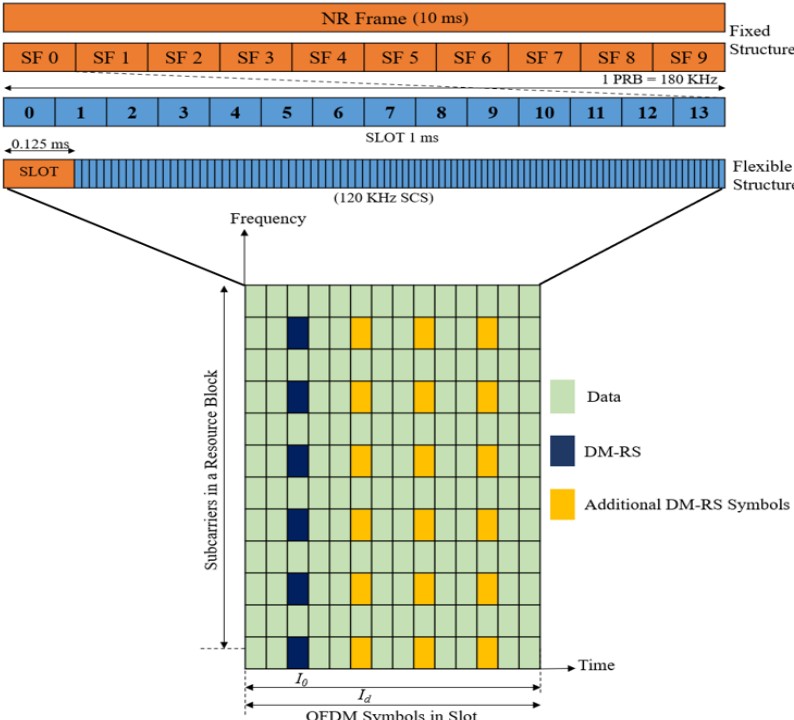

**Figure 4.** 5G-NR sub-frame with DM-RS symbol configuration to support high mobility.

The main objective of DM-RS is to estimate the channel coefficient for coherent detection. The design of DM-RS is closely connected to the channel characteristics. The channel fluctuates more rapidly in the frequency domain. In that situation, the DM-RS density within the frequency domain must be raised, resulting in a low channel coherence bandwidth [18]. This front-loaded DM-RS architecture is particularly useful for minimizing decoding delay when there is high mobility and the channel coherence time is longer than the period of the front-loaded DM-RS. On the other hand, if only the front-loaded DM-RS is assigned, link performance may degrade as UE speed increases because of the shorter channel coherence time. Figure 4 shows the DM-RS configuration used in practical channel estimation. The additional three DM-RS symbols are used to develop a denser pilot pattern to support high mobility.

## 5. Simulation Results and Discussion

The MATLAB 2022a platform is utilized to develop the link-level simulation in this paper. Table 2 lists the simulation parameters used for the 5G–NR PDSCH simulation performance analysis to support high mobility in the NR–NTN [1,9].

The performance of 5G-NR PDSCH in terms of BER and throughput versus the signal-to-noise ratio (SNR) is analyzed using link-level simulations in this paper. The simulation results illustrate how different numerologies, MCS levels, power delay profiles, MIMO, and channel estimation schemes affect the system's performance. The investigations are performing out in a multi-stream system to evaluate how the modulation order and propagation channel model influence the BER and throughput. In this analysis, different modulation schemes are simulated, including QPSK, 16-QAM, 64-QAM, and 256-QAM under NLOS, such as TDL-A and LOS, such as TDL-E for multipath fading channel model from which a single tap is simulated for analysis under single-path fading, such as Rayleigh and Rician fading distribution.

Figure 5 shows the BER performance, and Figure 6 illustrates the throughput performance of the NR-PDSCH LLS under single-path (1P) Rayleigh and Rician fading and multipath TDL-A and TDL-E as NLOS and LOS channel model, where the different MCS levels for SISO and MIMO schemes at stationary UE are simulated. The observed coding gain between SISO and MIMO ($2 \times 2$, $2 \times 4$, and $2 \times 8$) at BER of $10^{-2}$ under single-path Rayleigh fading distribution is 7, 8.5, and 11 dB in the case of the MCS level of QPSK. Under a Rician fading distribution, it is 6, 9, and 11dB, respectively. In contrast, under NLOS (TDL-A), the coding gain between SISO and MIMO cases is 9, 11, and 13 dB and 5, 9, and 11 dB under LOS (TDL-E) channel model, respectively. Similarly, the coding gain for other MCS levels, such as 16-QAM, 64-QAM, and 256-QAM, under NLOS and LOS propagation channels from Figure 5 is observed.

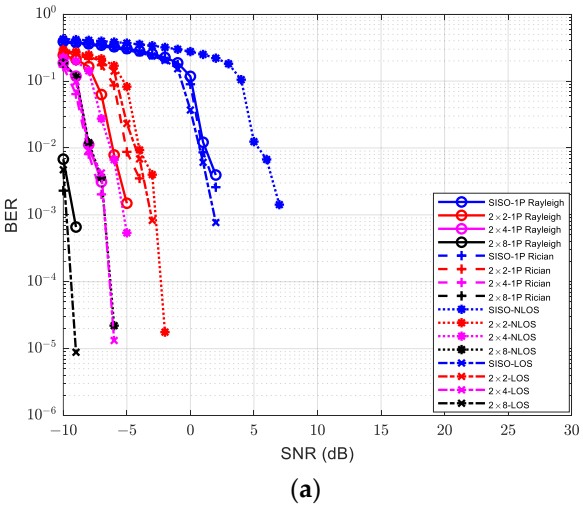

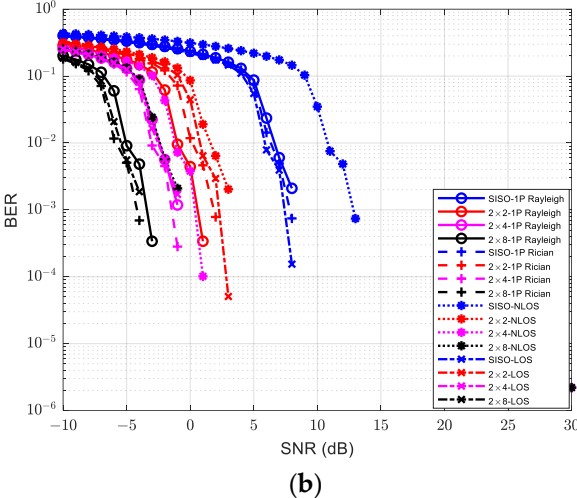

**(a)**      **(b)**

**Figure 5.** *Cont.*

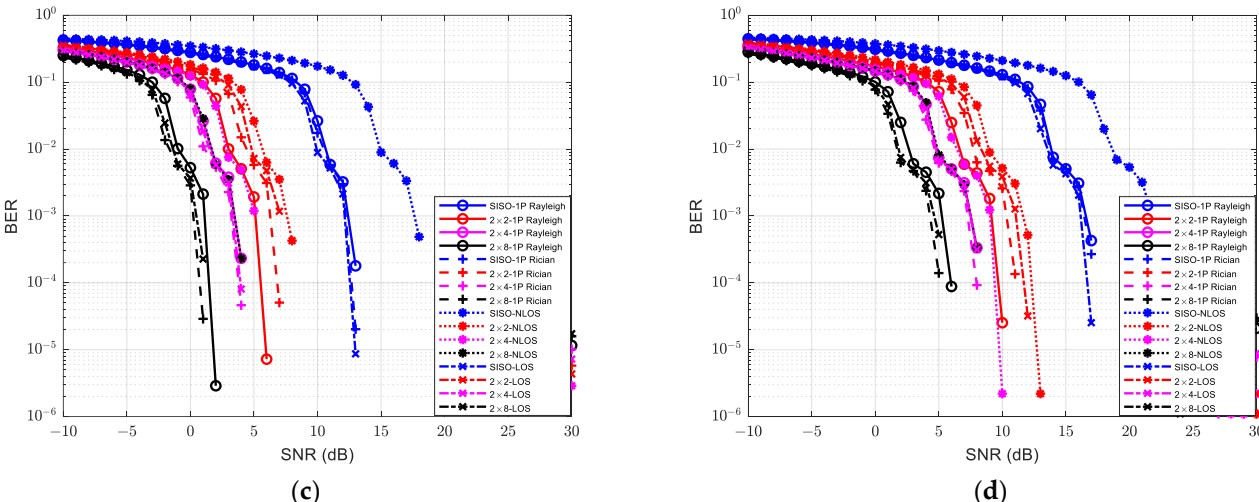

**Figure 5.** BER performance under various MCS levels for SCS 15 kHz at stationary UE: (**a**) QPSK, (**b**) 16-QAM, (**c**) 64-QAM, and (**d**) 256-QAM.

**Figure 6.** Throughput performance under various MCS levels for SCS 15 kHz at stationary UE: (**a**) QPSK, (**b**) 16-QAM, (**c**) 64-QAM, and (**d**) 256-QAM.

This paper analyzes the maximum achievable throughput of 5G-NR PDSCH under single and multipath fading channels designed by 3GPP. For NR-PDSCH, the maximum

achievable throughput is 30Mbps under 3GPP NR-NTN channel models. QPSK achieves high throughput in a low SNR, whereas 256-QAM achieves higher throughput in high SNR cases. According to Figure 6d, to obtain a maximum throughput of 30 Mbps, the 256-QAM under NLOS (TDL-A) channel model requires a minimum SNR(dB) of 18, 9, 6, and 5 dB for SISO and MIMO (2 × 2, 2 × 4, and 2 × 8) cases, respectively, and with MCS of QPSK under a TDL-E channel model, whereas 13, 8, 5, and 2 dB are required. This analysis shows that when the MIMO and 3D beamforming schemes are utilized, the maximum achievable throughput increases even at low SNR values.

The channel estimation provides a representation of the channel effects per resource element, and the equalizer uses that information to compensate for the distortion by the channel. In this paper, we assume channel knowledge in such a way that the pilot symbols are multiplexed with the data symbols at the transmit side within the resource grid, which represents the channel conditions between the transmit and receive antennas. The equalizer requires channel knowledge between the transmit layers and the receiving antennas that must apply precoding to the perfect channel estimation.

This paper analyzes the pilot-aided perfect and non-front-loaded DM-RS-based practical channel estimation for high numerology, which is SCS 120 kHz, for various MCS levels at 500 km/h in Figure 7. The results show that the system needs to be improved to support high mobility. Hence, in Figure 8, we have examined various numerologies at 500 km/h for 64-QAM under LOS and practical channel estimation. The high numerology of 5G-NR can support the mobility of UE in high-mobility scenarios because the transmit data can obtain sufficient space in the slot to avoid inter-symbol interference (ISI). Therefore, we propose utilizing 120 kHz of SCS to support high mobility in NTN. Figure 8 shows that the numerology 120 kHz achieves high link reliability and high throughput.

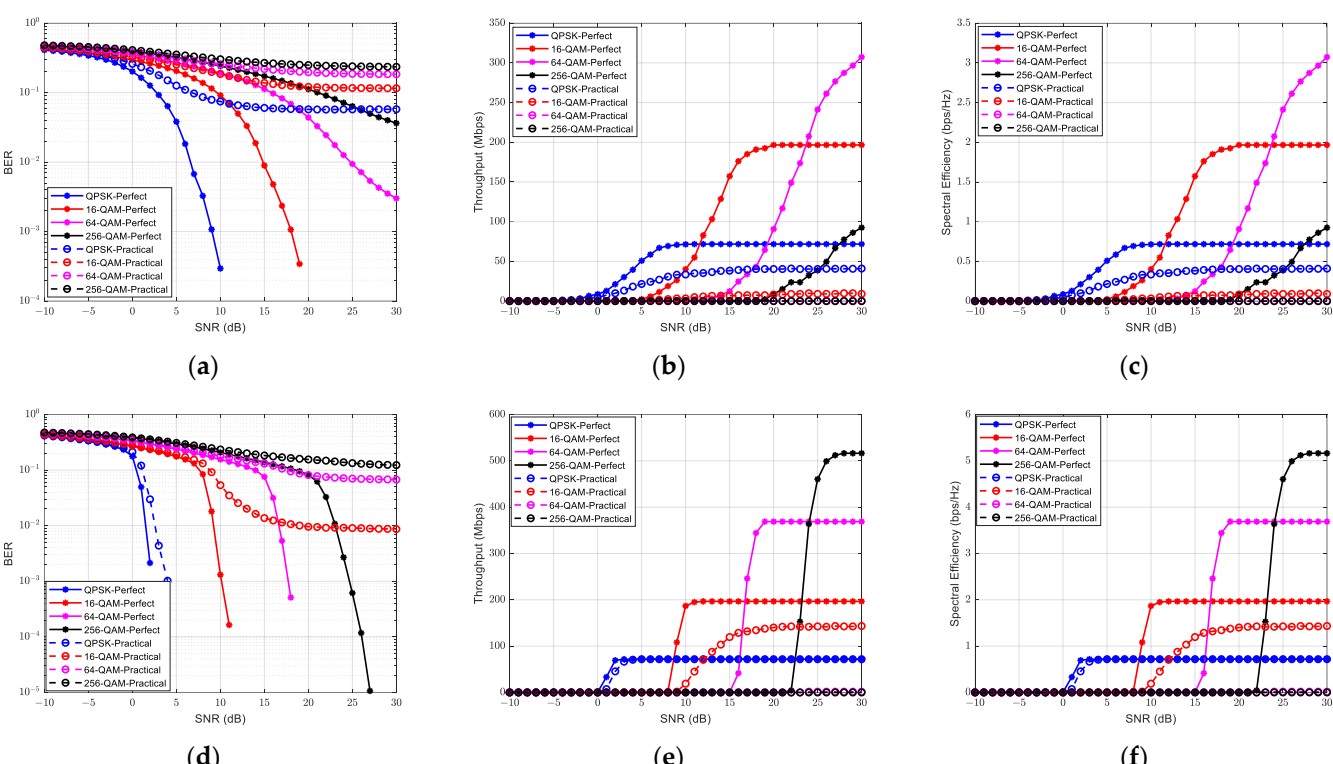

**Figure 7.** Link-level performance under pilot-aided perfect and DM-RS-based practical channel estimation at 500 km/h for TDL-A (NLOS): (**a**) BER, (**b**) throughput, (**c**) spectral efficiency; and for TDL-E (LOS): (**d**) BER, (**e**) throughput, (**f**) spectral efficiency.

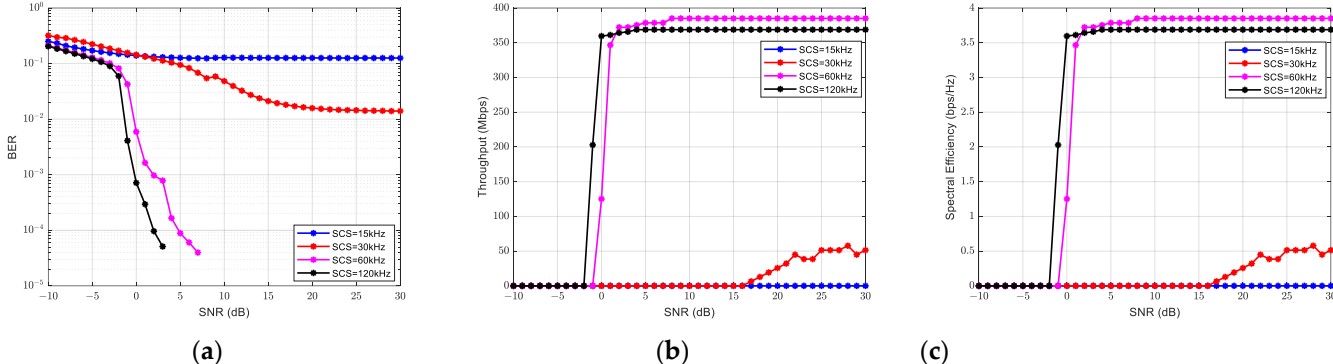

**Figure 8.** Link-level performance under various 5G-NR numerologies at 500 km/h: (**a**) BER, (**b**) throughput, (**c**) spectral efficiency.

The practical channel estimation uses the PDSCH DM-RS to estimate the channel conditions. It uses noise averaging and interpolation to obtain an estimate for all resource elements in the slot. This observation is because the DM-RS symbols are specified per layer, representing the channel conditions between the transmit layers and the receive antenna, including the effect of the MIMO precoding operation. Hence, the impact of various MIMO schemes is analyzed, as shown in Figure 9.

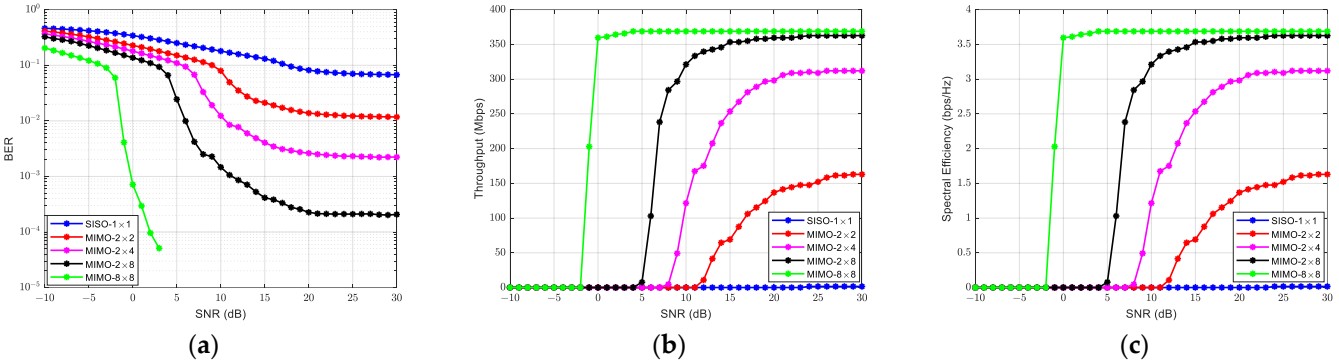

**Figure 9.** Link-level performance under various MIMO schemes at 500 km/h: (**a**) BER, (**b**) throughput, (**c**) spectral efficiency.

The various MIMO schemes are simulated for 64-QAM and 120 kHz SCS at 500 km/h under the TDL-A (LOS) channel model and practical channel estimation, as shown in Figure 9. A large number of transmit and receive antennae, such as (8 × 8) MIMO, have shown the highest link reliability. The maximum achievable throughput of 368.832 Mbps and spectral efficiency of 2.705 bps/Hz is achieved by the MIMO 8 × 8 scheme. The results in Figure 9 also show that utilizing the precoding-based MIMO beamforming is beneficial to support 500 km/h UE mobility in NTN. Therefore, we propose utilizing precoding-based MIMO (8 × 8) schemes to support high mobility in NTN.

The link-level performance of 5G-NR PDSCH is analyzed for NTN under the front-loaded DM-RS symbols with additional DM-RS symbols 0, 1, 2, and 3 in Figure 10. The simulation results show the improved link reliability with the three additional DM-RS symbols utilized in practical channel estimation, which is the denser symbol pattern of DM-RS symbols. High link reliability can be achieved despite sacrificing the throughput and spectral efficiency at UE mobility of 500 km/h. The front-loaded DM-RS enables fast decoding, which helps to improve link reliability. Therefore, this paper proposes 5G-NR physical layer elements, such as the SCS of 120 kHz, high MCS levels, such as 64-QAM or 256-QAM, and dense DM-RS symbol pattern-based channel estimation under MIMO schemes to support high mobility up to 500 km/h in NTN.

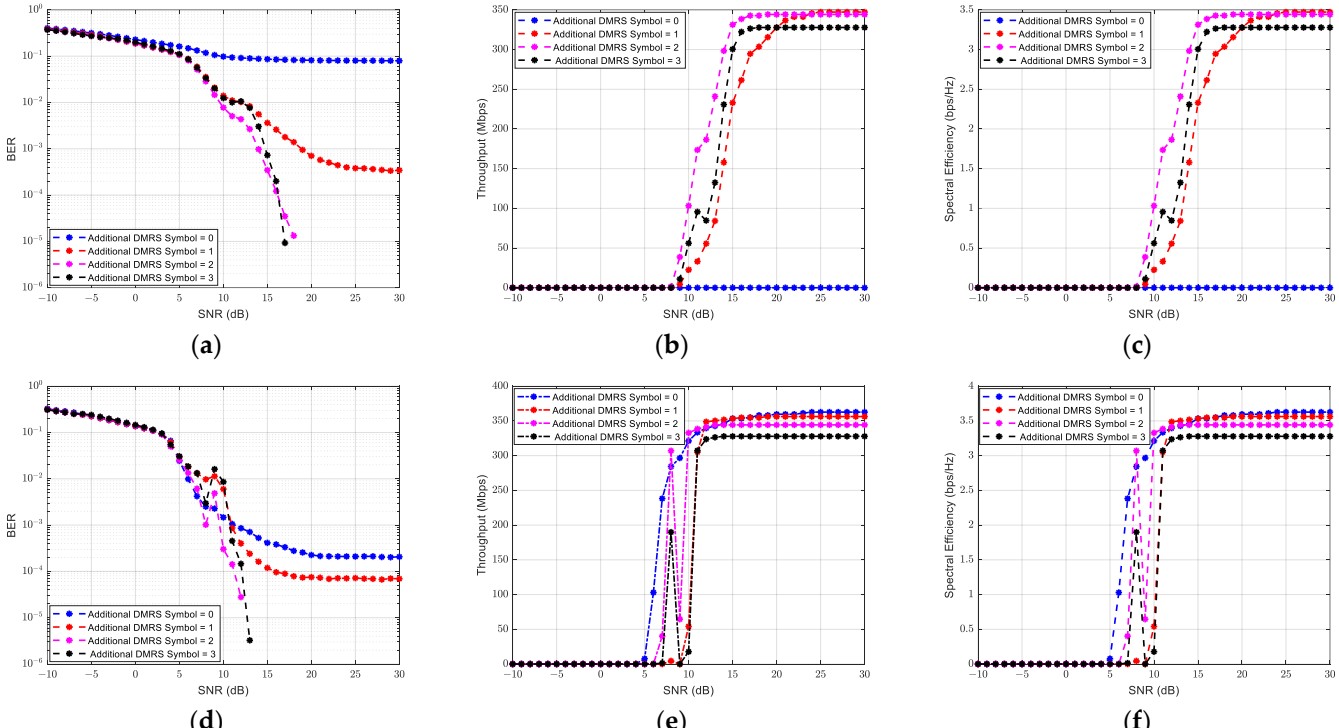

**Figure 10.** Link-level performance of front-loaded DM-RS-based channel estimation with additional DM-RS symbol for TDL-A (NLOS): (**a**) BER, (**b**) throughput, (**c**) spectral Efficiency; and for TDL-E (LOS): (**d**) BER, (**e**) throughput, (**f**) spectral efficiency.

## 6. Conclusions

This paper proposes the physical layer elements of 5G-NR for the evolution of non-terrestrial networks in 6G services based on the optimization of various 5G-NR numerologies, MCS levels, and MIMO schemes under pilot-aided perfect and DM-RS-based practical channel estimation schemes. The link-level simulations are carried out under 3GPP NR-NTN channel models, such as single-path Rayleigh, single-path Rician, multipath NLOS (TDL-A), and LOS (TDL-E) fading channels, at stationary UE and high UE mobility of 500 km/h. The link-level performance of 5G-NR PDSCH is analyzed in terms of BER, throughput, and spectral efficiency versus SNR (dB). The simulation results show that NR-PDSCH can achieve 30 Mbps throughput at stationary UE under high MCS levels, such as 256-QAM for SCS of 15 kHz, using a pilot-aided perfect channel estimation. The optimization of numerologies shows that the practical channel estimation requires a high SCS of 120 kHz to support 500 km/h. The front-loaded DM-RS can enable fast decoding. Hence, the DM-RS-based channel estimation with a denser DM-RS symbol pattern with a front-loaded configuration is proposed to support the high mobility along with the high SCS and MCS levels, such as 120 kHz and 64-QAM, respectively, under the MIMO (8 × 8) scheme, which provides the maximum achievable throughput of 368.832 Mbps and spectral efficiency of 3.68 bps/Hz at 500 km/h for the LOS (TDL-E) channel model. The utilization of SCS of 120 kHz, MCS 64-QAM, dense DM-RS symbol pattern-based channel estimation, and MIMO/3D-beamforming based on precoding shows improvement in the link reliability, which is proposed to utilize for NTN to support high mobility, which can also successfully enable transonic speed mobility in 6G-NTN services.

**Author Contributions:** Supervision and investigation, K.C.; Conceptualization and methodology, C.J.P.; writing—original draft preparation, C.J.P.; writing—review and editing, K.C. and C.J.P. All authors have read and agreed to the published version of the manuscript.

**Funding:** This work was supported by the Institute of Information & Communications Technology Planning & Evaluation (IITP) grant funded by the Korean government (MSIT) (No. 2021-0-00794, Development of 3D Spatial Mobile Communication Technology).

**Institutional Review Board Statement:** Not applicable.

**Informed Consent Statement:** Not applicable.

**Data Availability Statement:** Not applicable.

**Conflicts of Interest:** The authors declare no conflict of interest.

## Abbreviations

| | |
|---|---|
| 3GPP | Third-Generation Partnership Project |
| 5G | Fifth Generation |
| 6G | Sixth Generation |
| BER | Bit Error Rate |
| CDL | Clustered Delay Line |
| eNB | evolved Node B |
| gNB | 5G New Radio Node B |
| LLS | Link-Level Simulation |
| MCS | Modulation and Coding Schemes |
| MIMO | Multiple Input Multiple Output |
| NR | New Radio |
| NTN | Non-Terrestrial Networks |
| PDSCH | Physical Downlink Shared Channel |
| PHY | Physical Layer |
| PRB | Physical Resource Block |
| QoS | Quality of Service |
| RAN | Random Access Network |
| SNR | Signal-to-Noise Ratio |
| TDL | Tapped Delay Line |
| UAS | Unmanned Aerial System |
| UAV | Unmanned Aerial Vehicles |
| UE | User Equipment |

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
