# Peer review of "5G-NR Physical Layer-Based Solutions to Support High Mobility in 6G Non-Terrestrial Networks"

_drones, doi:10.3390/drones7030176_

Round 1
Reviewer 1 Report
I have no further comments or suggestions for the authors.
Reviewer 2 Report
This paper proposed the optimization of 5G-NR numerologies and the impact of various modulation and coding schemes (MCS), 3GPP NR-NTN channel models, and MIMO/beamforming schemes with link-level simulation under pilot-aided based perfect and DM-RS based practical channel estimation at stationary UE and high mobility of 14 500 km/h, respectively. Herein are some comments that can improve the quality of the article.
1- In Section 1 (Introduction), 3rd paragraph (contributions), I suggest summarizing the contributions and novelty of this study in bullet points.
2- I suggest summarizing Section 2 (Related Works ) in a table at the end of the section and highlighting the contribution of this study.
3- Some parts of the paper need to be meditated on linguistically
4- The 6G structure is not well defined, Authors are encouraged to clarify the 6G structure in an independent section
5- Should add a section titled "optimization problem model" to describe the optimization problem and constraints very well and the technique that you used to solve the optimization problem and why select this technique and show the validation in the results.
6-Table 1 needs to support by references
7-The resolution of the figures must be improved. the texts could not be read. You can put the figures vertically to enhance the quality of reading.
Reviewer 3 Report
The article is confused about dealing 5G and 6G. The proposal does not consider some aspect of high-speed scenarios and does not specify which channel model.
Reviewer 4 Report
The article presented by the authors develops a link-level simulation of the 5G-NR physical downlink shared channel (PDSCH) under the 3GPP NR-NTN tapped delay line (TDL) channel model to support UE mobility up to 500 km/h. The bit error rate (BER), maximum achievable throughput (Mbps), and spectral efficiency (bps/Hz) are analyzed for the 5G-NR-based potential elements to utilize in the evolution of NTN. Furthermore, the denser DM-RS symbol pattern is proposed to utilize for channel estimation to support high mobility, as simulation results prove their capability of fast decoding while using the front-loaded symbol structure.
1. The article is well written, apart from some English mistakes. a second reading is needed to improve the article.
2. I find that the introduction is too limited and it is difficult to see the contribution. please add a section and indicate your motivations. moreover, it is preferable to add a comparison table between important articles in the literature and your contribution.
3. In figure 1 each vector means a channel please indicate each vector according to the chosen colour.
4. The figures in your simulations are not clear. it is recommended to review the figures.
5. Not all your results are justified. please try to give more explanation on your results. on the other hand, it is important to compare your results with other methods or articles.
6. Therefore, choosing a very high subcarrier spacing may result in high CP overhead. Can you explain this ?
7. all your results are based on the SNR. how you calculate the SNR. an equation is needed. what is the number of users used in your simulation.
8. Same for the flow rate how you calculate it ?
9. Each block in figure 3 should be explained briefly. how your propagation model is calculated. A lot of information is missing.
10. Please add this reference with reference 3 and 4
M. A. Ouamri, R. Alkanhel, C. Gueguen, M. A. Alohali and S. S. M. Ghoneim, "Modeling and analysis of uav-assisted mobile network with imperfect beam alignment," Computers, Materials & Continua, vol. 74, no.1, pp. 453–467, 2023.
J. Li, Y. Xiong, J. She and M. Wu, "A Path Planning Method for Sweep Coverage With Multiple UAVs," in IEEE Internet of Things Journal, vol. 7, no. 9, pp. 8967-8978, Sept. 2020
Round 2
Reviewer 2 Report
The authors have addressed the raised comments. However, the resolution of the figures very low; could not read the text. I think the number of pages is not an issue with the MDPI publisher, the important is the quality.
Reviewer 3 Report
Many thanks to the authors for trying to address my concern. Although the explanation has improved, I still have doubts about the channel model they have. They have used the standard in a very general way, without exactly a solid argument for why they chose that model.
- In the following reference [1], I have seen that they offer a list of models for applications including the proposal in this work with more arguments. Could the authors explain among the following models which is the most adapted model to the one they present?
[1] V. M. Baeza, E. Lagunas, H. Al-Hraishawi and S. Chatzinotas, "An Overview of Channel Models for NGSO Satellites," 2022 IEEE 96th Vehicular Technology Conference (VTC2022-Fall), London, United Kingdom, 2022, pp. 1-6.
Finally, I still think that there is a mix between 5G and 6G confusing. It would have been clearer to opt for one of the 2 technologies. However, I understand that now everything can still be 6G and thus have greater visibility.
With the clarification of the first two points in the introduction at least, I am happy to recommend this work.
Reviewer 4 Report
The author improved the paper
I acczpt this work
